# Blue light-induced LOV domain dimerization enhances the affinity of Aureochrome 1a for its target DNA sequence

**Udo Heintz\*, Ilme Schlichting**

Department of Biomolecular Mechanisms, Max Planck Institute for Medical Research, Heidelberg, Germany

**Abstract** The design of synthetic optogenetic tools that allow precise spatiotemporal control of biological processes previously inaccessible to optogenetic control has developed rapidly over the last years. Rational design of such tools requires detailed knowledge of allosteric light signaling in natural photoreceptors. To understand allosteric communication between sensor and effector domains, characterization of all relevant signaling states is required. Here, we describe the mechanism of light-dependent DNA binding of the light-oxygen-voltage (LOV) transcription factor Aureochrome 1a from *Phaeodactylum tricornutum (Pt*Au1a) and present crystal structures of a dark state LOV monomer and a fully light-adapted LOV dimer. In combination with hydrogen/deuterium-exchange, solution scattering data and DNA-binding experiments, our studies reveal a light-sensitive interaction between the LOV and basic region leucine zipper DNA-binding domain that together with LOV dimerization results in modulation of the DNA affinity of *Pt*Au1a. We discuss the implications of these results for the design of synthetic LOV-based photosensors with application in optogenetics.

**\*For correspondence:** Udo. Heintz@mpimf-heidelberg.mpg. de

**Competing interests:** The authors declare that no competing interests exist.

## Introduction

Light-sensing is essential for the survival of organisms from all kingdoms of life and plays an important role in their adaptation to different habitats. Prokaryotes, higher plants, fungi, animals and algae use light-sensing systems that encompass a variety of sensory photoreceptors that respond to different wavelengths of light. Recently, a new type of blue light photoreceptor termed Aureochrome (Aureo) was discovered in the photosynthetic stramenopile alga *Vaucheria frigida* that has been suggested to function as blue light-regulated transcription factor. Originally, two Aureo homologs, named Aureo 1 and 2, were identified, but only Aureo1 was shown to bind DNA in a light-dependent manner (*Takahashi et al., 2007*). Since the discovery of the first Aureos, several orthologs from other stramenopile algae such as *Ochromonas danica*, *Fucus distichus*, *Saccharina japonica* and *Phaeodactylum tricornutum* have been identified (*Ishikawa et al., 2009*; *Deng et al., 2014*; *Schellenberger Costa et al., 2013*). The diatom *P. tricornutum* has four genes encoding aureochromes: three orthologs of type 1 (*Pt*Au1a, b and c) and one of type 2 (*Depauw et al., 2012*). Only *Pt*Au1a has been functionally characterized so far and is shown to be involved in light-dependent mitosis regulation (*Huysman et al., 2013*) and repress high-light acclimation (*Schellenberger Costa et al., 2013*).

Aureos typically consist of an N-terminal domain with unknown function, a basic region leucine zipper (bZIP) DNA-binding domain, and a C-terminal light-oxygen-voltage (LOV) sensing domain. LOV domains are a subgroup of the Per-Arnt-Sim (PAS) superfamily that sense blue light using a

**eLife digest** The ability to react to sunlight is important for the survival of a wide range of lifeforms. Many organisms, including humans, plants, bacteria and algae, sense light using specialized proteins called photoreceptors. These proteins are able to translate the information transported by light into various biological activities.

The structure of a photoreceptor can be broken down into different parts, each with a specialized role. For example, the light-sensing region of a photoreceptor typically binds to small molecules called chromophores that are able to absorb light. This light absorption causes changes in the photoreceptor that are ultimately transmitted to a part of the protein that can bind to DNA or perform some other type of biological activity. This activity triggers further processes that build up to the organism's reaction to the incoming light.

Aureochromes are photoreceptors that detect blue light and are found in algae. The light-sensing and DNA-binding parts of aureochromes are arranged in a different way to the arrangement seen in most related photoreceptors. This raises questions about how the light signal is transmitted to the DNA-binding part of the protein and how this affects the DNA binding of aureochromes.

By using a combination of biophysical and structural methods, Heintz and Schlichting now provide detailed information about the structural changes that blue light causes in the Aureochrome 1a photoreceptor found in the algae *Phaeodactylum tricornutum*. This shows that when exposed to light, the light-sensing part of the photoreceptor, called LOV domain, detaches from the DNA binding part and binds to the LOV region of a second molecule. This helps the protein to bind to DNA.

Recently, synthetic photoreceptors have been engineered that use the light-sensing part of aureochromes. Therefore, as well as contributing to the fundamental understanding of light signaling in photoreceptors, Heintz and Schlichting's findings can be used to help develop light-controllable artificial proteins for use in research, medicine or industry.

noncovalently bound flavin cofactor (*Zoltowski and Gardner, 2011*; *Herrou and Crosson, 2011*; *Losi and Gartner, 2012*; *Conrad et al., 2014*). Photon absorption of the flavin results in formation of a flavin-C4(a)-cysteinyl adduct with a conserved cysteine residue (*Salomon et al., 2000*), which initiates a cascade of structural rearrangements within the LOV core that are propagated to the domain boundaries. LOV domains can be found as isolated entities, but are often part of multidomain proteins where they are coupled to a variety of different effector domains. The effector-sensor topology observed in Aureos differs from the domain topology found in most other LOV photoreceptors where the sensory LOV domain is located N-terminally to the effector domain. This rare domain topology raises the question of how light signaling is achieved in Aureos compared with other LOV proteins.

Recent biochemical and spectroscopic experiments on *V. frigida* Aureo1 (*Vf*Au1) and *Pt*Au1a led to the hypothesis that DNA binding of Aureos might be influenced by light-induced LOV domain dimerization and that structural changes of the N– (A´α) and C-terminal (Jα) helices flanking the LOV core play a key role in this process (*Toyooka et al., 2011*; *Herman et al., 2013*; *Mitra et al., 2012*; *Herman and Kottke, 2015*). This hypothesis was supported to some extent by the crystal structure of the *Vf*Au1 LOV domain that showed an unexpected dimeric arrangement (*Mitra et al., 2012*). However, this structure was determined from crystals grown in the dark and it remains unclear whether the observed dimeric LOV arrangement represents the biologically relevant light state dimer. To obtain insights into structural rearrangements within VfAu1 LOV upon light activation, dark state crystals were illuminated to study light-induced conformational changes (*Mitra et al., 2012*). However, crystal lattice constraints can prevent large conformational changes that limit this approach. Therefore, the mechanism of light-induced LOV dimerization and its consequences on DNA binding in Aureos remain unclear.

Here, we present crystal structures of a fully light-adapted LOV dimer as well as of a dark state LOV monomer of *Pt*Au1a. We combine these results with hydrogen/deuterium-exchange coupled to mass spectrometry (HDX-MS) and small-angle X-ray scattering (SAXS) experiments of full-length *Pt*Au1a and of a truncated construct that lacks the N-terminal domain, respectively. Together with

functional studies, this integrative structural approach enabled us to establish a model for light signaling in Aureos where, in the dark, the LOV domain directly interacts with the bZIP domain and thereby impedes its DNA binding function. Illumination with blue light triggers intramolecular bZIP–LOV dissociation and subsequent LOV dimerization, thus enhancing the affinity of $PtAu1a$ for its target DNA sequence. Together, these results provide insight into the molecular mechanism of Aureo function and implicate a new model of light-dependent gene regulation by Aureos in stramenopiles. In addition, they offer new design strategies for synthetic Aureo–LOV based photosensors for applications in optogenetics.

## Results

### Dark state recovery of $PtAu1a_{full}$, $PtAu1a_{bZIP-LOV}$ and $PtAu1a_{LOV}$

To understand how blue light-sensing of the LOV domain influences $PtAu1a$ DNA binding, we used the full-length protein ($PtAu1a_{full}$) and additionally generated N- and C-terminally truncated $PtAu1a$ variants encompassing the bZIP and LOV domain ($PtAu1a_{bZIP-LOV}$), the bZIP domain ($PtAu1a_{bZIP}$) as well as the LOV domain ($PtAu1a_{LOV}$) containing its N- and C-terminal α-helical extensions A′α (in the context of PAS domains often referred to as N-cap) and Jα, respectively (*Figure 1a*). UV-vis spectra of dark adapted LOV domain containing $PtAu1a$ variants show the typical signature of an oxidized flavin mononucleotide (FMN) chromophore with a main absorption maximum at 448 nm and several subsidiary peaks (*Figure 1b*). Upon light activation, the intensities of these absorption bands decrease and a new maximum appears at 390 nm indicating FMN-cysteine adduct formation. Investigation of the dark state recovery kinetics of $PtAu1a_{full}$, $PtAu1a_{bZIP-LOV}$ and $PtAu1a_{LOV}$ yielded traces that were analyzed by fitting an exponential function (*Figure 1c*). For $PtAu1a_{full}$, $PtAu1a_{bZIP-LOV}$ and $PtAu1a_{LOV}$, time constants of 826 ± 19 s, 811 ± 4 s and 1500 ± 7 s were determined, respectively, indicating that the presence of the bZIP DNA binding domain accelerates the recovery kinetics of the LOV domain about 1.8-fold.

### Blue light illumination induces dimerization of $PtAu1a_{LOV}$, but not of $PtAu1a_{full}$ and $PtAu1a_{bZIP-LOV}$

To investigate the effect of blue light illumination on the oligomerization of $PtAu1a_{full}$, $PtAu1a_{bZIP-LOV}$, $PtAu1a_{LOV}$ we performed size-exclusion chromatography coupled to multi-angle light scattering (MALS) in the dark and under continuous blue light illumination. Quantification of the average molar mass of $PtAu1a_{full}$ (*Figure 2a*) and $PtAu1a_{bZIP-LOV}$ (*Figure 2b*) in the dark and light state yielded

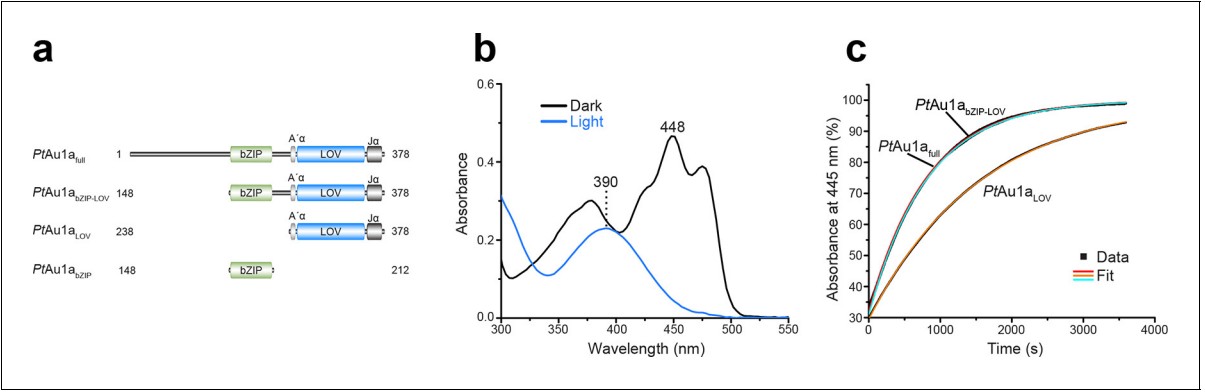

**Figure 1.** Absorption spectra of $PtAu1a_{full}$ and dark state recovery kinetics of the LOV domain-containing $PtAu1a$ variants. (**a**) Schematic representation of the $PtAu1a$ constructs used in this study. (**b**) Absorption spectrum of $PtAu1a_{full}$ in the dark (black) and after illumination with blue light (blue). In the dark, the typical signature of an oxidized FMN chromophore can be detected. Upon light activation, these maxima decrease and a new absorption band appears at 390 nm, indicating FMN-cysteine adduct formation. (**c**) Recovery kinetics of the absorbance at 445 nm of $PtAu1a_{full}$, $PtAu1a_{bZIP-LOV}$ and $PtAu1a_{LOV}$ after light-activation. The red ($PtAu1a_{full}$), cyan ($PtAu1a_{bZIP-LOV}$) and orange ($PtAu1a_{LOV}$) lines in the plot represent an exponential fit to the data (black squares). Measurements were performed at a protein concentration of 20 μM and time constants represent the mean of three independent measurements. LOV, light-oxygen-voltage; FMN, flavin mononucleotide.

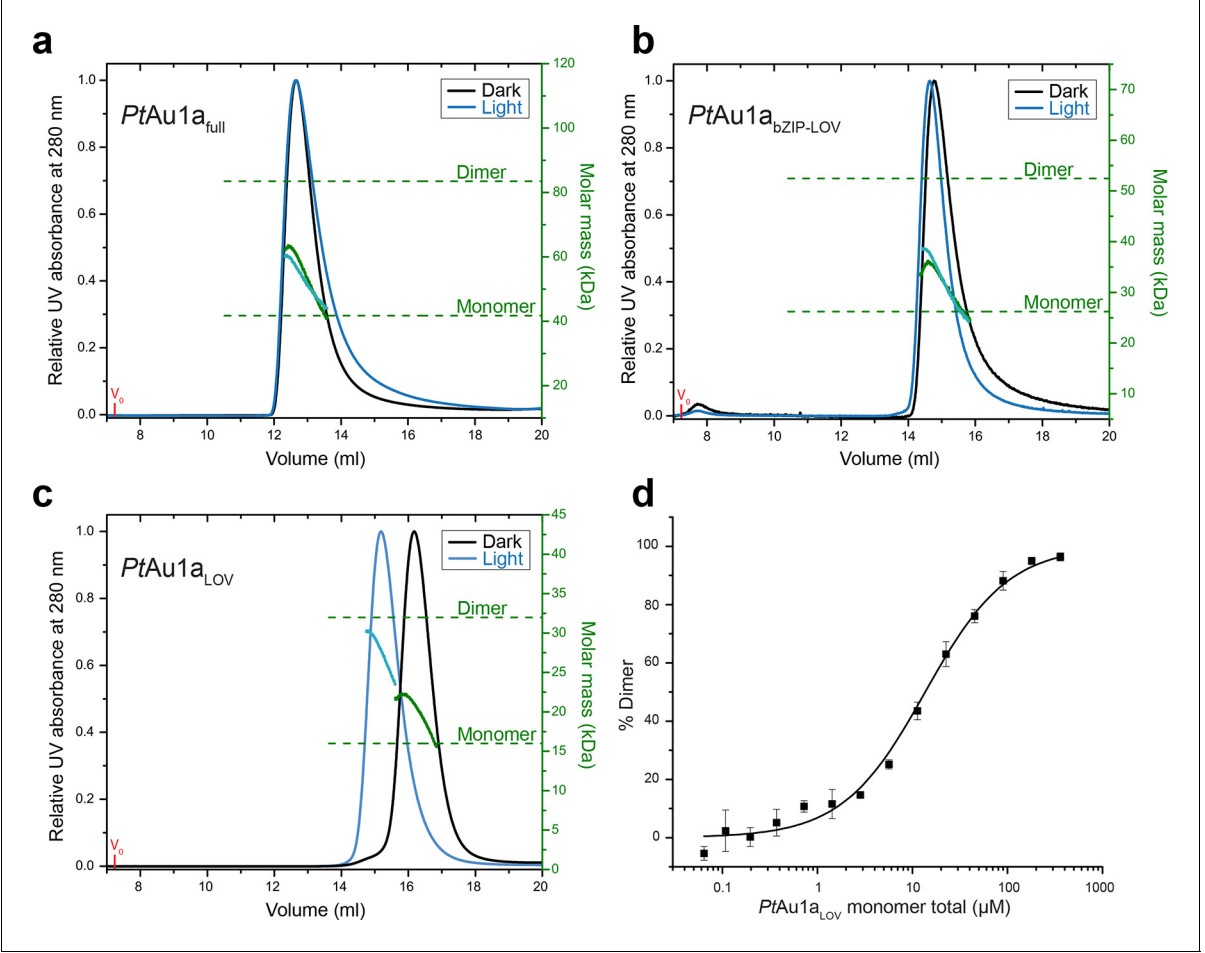

**Figure 2.** Domains involved in *Pt*Au1a dimerization. Normalized MALS detection of *Pt*Au1a_full (a), *Pt*Au1a_bZIP-LOV (b) and *Pt*Au1a_LOV (c) fractionated by size-exclusion chromatography in the dark (black traces) and light (blue traces). The MALS-derived molar-mass signals are shown in green (dark runs) and blue–green (light runs). Additional experiments performed for *Pt*Au1a_full and *Pt*Au1a_bZIP-LOV in the light at varying protein concentrations are shown in *Figure 2—figure supplement 1*. (d) Quantification of the monomer-dimer equilibrium of *Pt*Au1a_LOV in the dark by MST. Error bars represent the standard deviation of three individual experiments. MALS, multi-angle light scattering; MST, microscale thermophoresis.

The following figure supplement is available for figure 2:

**Figure supplement 1.** Concentration dependent elution profiles of *Pt*Au1a_full and *Pt*Au1a_bZIP-LOV in the light.

similar values of 55.1, 53.3, 31.7 and 33.7 kDa, respectively, which are between the theoretically expected molar masses of dimers (83.6 kDa for *Pt*Au1a_full and 52.6 for *Pt*Au1a_bZIP-LOV) and monomers (41.8 kDa for *Pt*Au1a_full and 26.3 kDa for *Pt*Au1a_bZIP-LOV). Peak tailing and a continuous decrease of the molar mass signal in the dark and light state of *Pt*Au1a_full and *Pt*Au1a_bZIP-LOV suggested an equilibrium between dimers and monomers irrespective of the light conditions. Illumination-induced small differences in the elution volumes, indicating conformational changes of both protein variants have occurred as also reported for *Vf*Au1 (*Toyooka et al., 2011*; *Hisatomi et al., 2013*). In contrast to *Pt*Au1a_full and *Pt*Au1a_bZIP-LOV, light activation of *Pt*Au1a_LOV shifts the oligomerization state from monomers to dimers (*Figure 2c*). The change in oligomerization state is reflected by a decrease of the elution volume from 16.2 ml (dark) to 15.2 ml (light) and an increase of the calculated average molar mass from 20 kDa (dark) to 28 kDa (light). Quantification of the monomer–dimer transition using microscale thermophoresis (MST) revealed a $K_d$ of $13.6 \pm 1.4$ µM for the dark adapted protein (*Figure 2d*) and pre-illumination of *Pt*Au1a_LOV increased the dimerization ability as expected from the MALS measurements (data not shown). However, it was not possible to determine a reliable $K_d$ value for the monomer–dimer transition of light-activated *Pt*Au1a_LOV, as

continuous illumination of the protein is not possible during the measurements. Together, our results are in line with previous reports on the oligomerization states of the LOV domain of *Pt*Au1a (*Herman et al., 2013*; *Herman and Kottke, 2015*) as well as truncated and full-length variants of *Vf*Au1 in their dark and light states (*Toyooka et al., 2011*; *Hisatomi et al., 2013*). It was recently reported that *Vf*Au1 dimerizes in a light-dependent manner and that the redox potential influences the oligomerization state by formation of disulfide bonds between the bZIP domains and the bZIP–LOV linker regions (*Hisatomi et al., 2014*). We did not observe such light-dependent oligomerization in our experiments and can also rule out an influence of the redox potential on the oligomerization state of *Pt*Au1a, as *Pt*Au1a does not possess cysteine residues outside the LOV domain.

## Blue light illumination enhances the affinity of *Pt*Au1a for its target DNA sequence

To characterize the effect of blue light on the DNA binding properties of *Pt*Au1a$_{full}$ we performed electrophoretic mobility shift assays (EMSAs) in the dark as well as under continuous blue light illumination and measured the binding of *Pt*Au1a to a 24-base pair (bp) DNA fragment of the *diatom-specific cyclin 2 (dsCYC2*, GenBank XM_002179247) promoter sequence of *P. tricornutum* containing the TGACGT binding motif reported for *Vf*Au1 (*Takahashi et al., 2007*) (*Figure 3a,b*). Semi-quantitative data evaluation using the Hill equation revealed an effective concentration for 50% response (EC$_{50}$) of 860 nM in the dark and a Hill coefficient of 1.35 (*Figure 3—figure supplement 1*). Illumination with blue light results in a decrease of the EC$_{50}$ to 90 nM (Hill coefficient of 1.65), revealing a 9.6-fold higher affinity of *Pt*Au1a$_{full}$ to DNA in its light compared with its dark state. To verify sequence-specific DNA binding of *Pt*Au1a$_{full}$, we performed the same experiments with a 24-bp DNA fragment lacking the *Pt*Au1a target sequence (*Figure 3—figure supplement 2*). In the dark as well as light experiments, *Pt*Au1a$_{full}$ displayed no or only weak binding to the DNA probe lacking the target sequence, confirming sequence-specific DNA binding of *Pt*Au1a$_{full}$. The presence of MgCl$_2$ in the EMSA experiments is essential for *Pt*Au1a$_{full}$ sequence specificity, as also described for other bZIP transcription factors (*Moll, 2002*). In the absence of MgCl$_2$, sequence specificity and light-dependence of DNA binding are negligible (*Figure 3—figure supplement 3*).

## *Pt*Au1a$_{LOV}$ light and dark state structures

To investigate the underlying molecular mechanism for light-regulated gene transcription by *Pt*Au1a and the effect of LOV dimerization, we solved the crystal structure of *Pt*Au1a$_{LOV}$ in its dark and light state (*Figure 4a and b* and *Table 1*). The dark state structure revealed a LOV monomer that adopts a typical PAS fold consisting of a five-stranded antiparallel β-sheet flanked by several helices. The LOV core forms the chromophore binding pocket and closely resembles the structure of *Vf*Au1 LOV (*Mitra et al., 2012*) (root-mean-square deviation (r.m.s.d) between 0.42 and 0.58 Å for 101 C$^\alpha$ atoms and molecules A and B *Pt*Au1a and A,B,C,D,E,F for *Vf*Au1a). The LOV core is flanked at the N- and C-termini by prominent α-helical extensions denoted A´α and Jα, respectively. As observed for *Vf*Au1 LOV (*Mitra et al., 2012*), the C-terminal Jα helix partially folds back onto the surface of the β-sheet and interacts with the LOV core via hydrogen bonds between the conserved residue Gln365 and the carbonyl and amine group of Cys316 as well as the side chains of Tyr357 and Gln330. A´α forms an amphipathic three-turn helix and interacts with the LOV core through a highly conserved 4 amino acid linker with Ala248, Glu249, Glu250 and Gln251 in the hinge region. In addition to Jα, A´α also folds back across the surface of the β-sheet and covers a large hydrophobic patch (*Figure 4c*). The chromophore binding pocket is mainly formed by hydrophobic residues that stabilize the FMN chromophore together with Gln350, Gln291, Asn319, Asn329, which form hydrogen bonds with the heteroatoms of the isoalloxazine ring. FMN is additionally stabilized by Arg304 and Arg288, which interact with the phosphate group of the ribityl chain. The conserved photoreactive Cys287, which forms a covalent bond with FMN upon illumination, is located in the Eα helix on the opposite site of the core β-sheet.

Light-induced dimerization of *Pt*Au1a$_{LOV}$ requires tertiary and quaternary structural rearrangements that cannot be induced by illumination of dark state crystals due to crystal lattice restraints. Therefore, to detect the full extent of light-induced structural changes, crystals of the light state need to be grown under continuous blue light illumination of the setup. The increase in structural dynamics and aggregation tendency upon illumination of most photoreceptors as well as potential

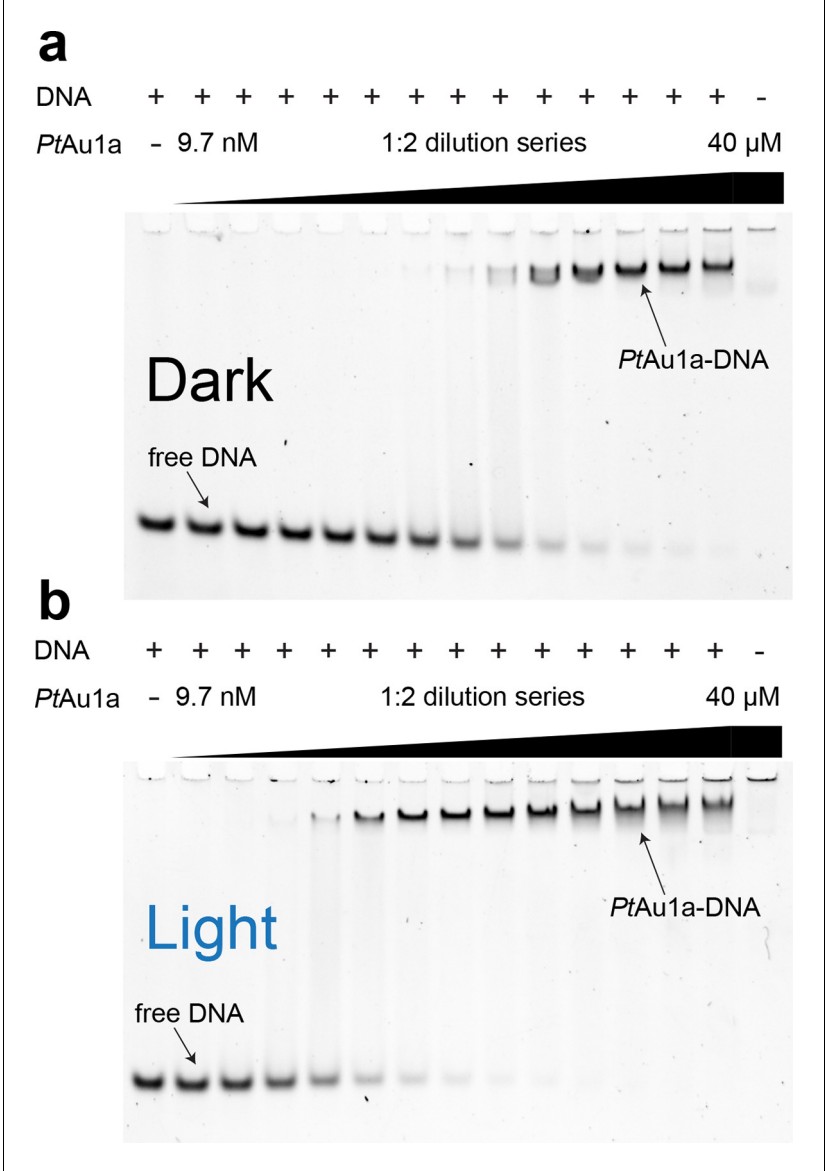

**Figure 3.** Blue light illumination enhances DNA binding of $PtAu1a_{full}$ to its target DNA sequence. EMSAs of $PtAu1a_{full}$ under dark (**a**) and light (**b**) conditions in the presence of 50 nM $dsCYC2$ promoter DNA. Quantification of the gels in *Figure 3—figure Supplement 1*. EMSAs, electrophoretic mobility shift assays.

The following figure supplements are available for figure 3:

**Figure supplement 1.** DNA binding curves of $PtAu1a_{full}$ under dark and light conditions.

**Figure supplement 2.** $PtAu1a_{full}$ binds DNA in a sequence-specific manner.

**Figure supplement 3.** $MgCl_2$ is essential for sequence-specific DNA binding of $PtAu1a_{full}$.

photobleaching makes crystallization of photoreceptors difficult and has so far only been achieved for the single LOV domain proteins VIVID (*Vaidya et al., 2011*) and PpsB1 (*Circolone et al., 2012*) as well as for a truncated phytochrome construct (*Takala et al., 2014*). These photoreceptors revert slowly back into their dark conformation or, as in the case of VIVID, have been modified to do so. To obtain the structure of a fully light-adapted $PtAu1a_{LOV}$ dimer, we set up crystallization screens of wildtype $PtAu1a_{LOV}$ under continuous blue light illumination and obtained colorless crystals

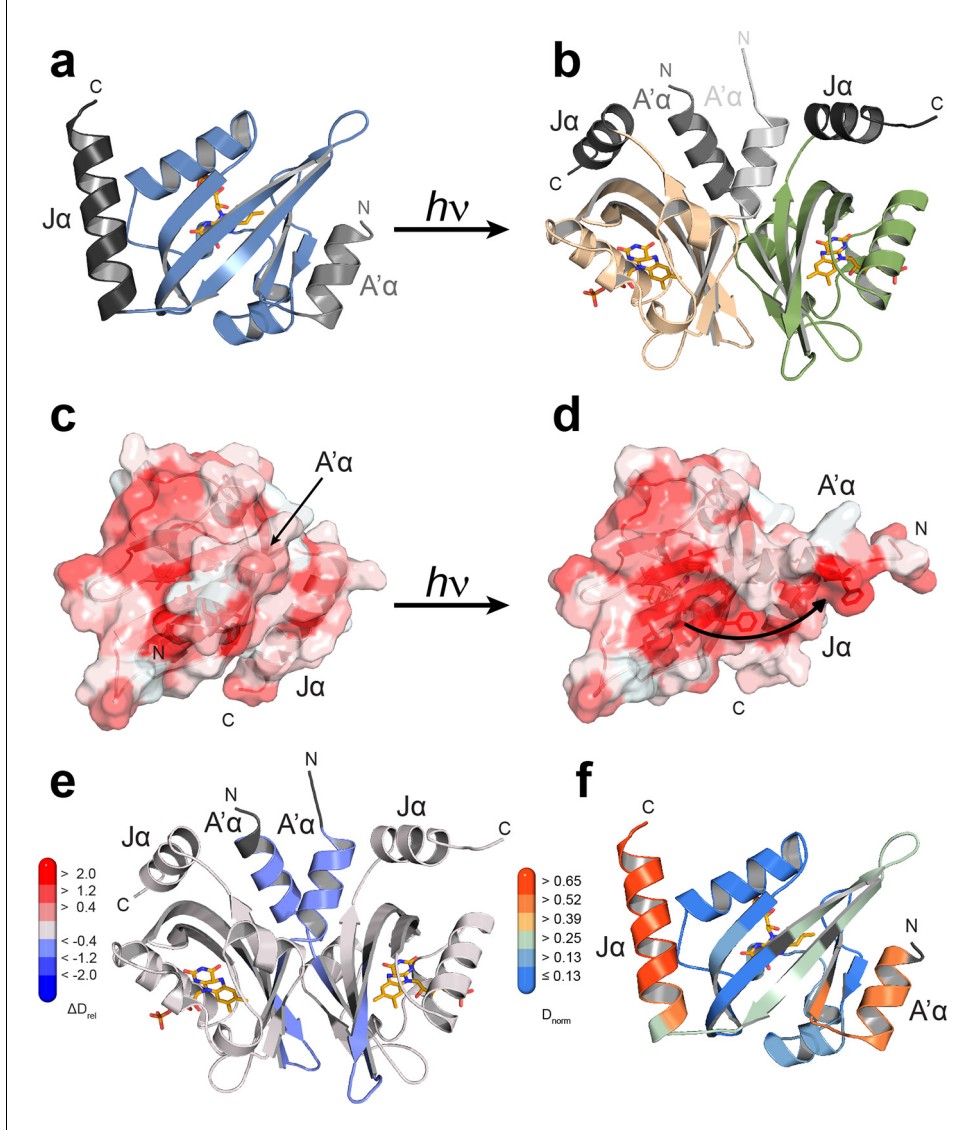

**Figure 4.** Structural characterization of $PtAu1a_{LOV}$ in its dark and light state. (**a**) Crystal structure of the $PtAu1a_{LOV}$ dark state monomer with the N- and C-terminal A'α and Jα helices flanking the LOV core colored in light gray and dark gray, respectively. (**b**) Blue light illumination induces formation of a parallel $PtAu1a_{LOV}$ dimer. (**c**) In the dark, A'α covers the hydrophobic dimerization site on the LOV β-sheet. (**d**) Illumination results in a release of A'α from the LOV β-sheet and exposes the dimerization site. The $PtAu1a_{LOV}$ molecules in (c and d) are colored according to the Eisenberg hydrophobicity scale (***Eisenberg, et al., 1984***). Reddish regions correspond to high and white regions to low hydrophobicity. (**e**) The $PtAu1a_{LOV}$ light state dimer colored according to differences in deuterium incorporation in the dark and light state after 10 s of labeling. Shades of red and blue correspond to regions with increased and decreased deuterium uptake in the light, respectively. A peptide map that shows the differences in relative deuteration of dark and light experiments for all time points is shown in (***Figure 4—figure supplement 3***). All evaluated peptides for $PtAu1a_{LOV}$ and their individual deuteration plots are shown in (***Figure 4—figure supplement 4***). (**f**) $PtAu1a_{LOV}$ dark state monomer colored according to deuterium incorporation in the dark after 10 s labeling. Elements in (**e**) and (**f**) colored in dark gray represent regions that are not covered by peptides generated by pepsin digestion. Since rapid back-exchange of the two N-terminal residues prevents precise measurement of deuterium incorporation, these residues of all peptides are shown in dark gray, if not covered by an overlapping peptide. LOV, light-oxygen-voltage.

The following figure supplements are available for figure 4:

**Figure supplement 1.** Light-induced structural changes of $PtAu1a_{LOV}$ (protomer A).

*Figure 4 continued on next page*

*Figure 4 continued*

**Figure supplement 2.** Interdomain interactions and crystal lattice contacts.

**Figure supplement 3.** Effect of illumination on *Pt*Au1a$_{LOV}$.

**Figure supplement 4.** Overview of all 39 *Pt*Au1a$_{LOV}$ peptides evaluated during HDX-MS analysis.

overnight. Their lack of color is indicative of formation of the flavin-C4(a)-cysteinyl adduct. As expected from the MALS measurements, light-activated *Pt*Au1a$_{LOV}$ crystallized as a dimer that features an assembly similar to other N-cap-comprising PAS dimers (*Figure 4b*) (*Heintz et al., 2014*; *Ayers and Moffat, 2008*). The two *Pt*Au1a$_{LOV}$ protomers are related by a 2-fold non-crystallographic symmetry that does not apply to the A′α helices. The observed parallel dimer arrangement differs significantly from the antiparallel dimer arrangement observed for the dark state structure of *Vf*Au1 LOV (*Mitra et al., 2012*). Analysis of the contact area using the *PISA* web server revealed a buried surface area (BSA) of 1524 Å$^2$, which is similar to that observed for the VIVID light state dimer (1342 Å$^2$), supporting the observed dimer arrangement as biologically relevant. The main dimerization interface is formed by a network of hydrophobic residues on the core β-sheet and A′α. Superposition of the dark and light state structures not only reveals a variety of light-induced side chain rearrangements (*Figure 4—figure supplement 1*), but also a significant change of the position of the A′α helix that is released from the hydrophobic patch on the LOV core, thus exposing the dimerization site (*Figure 4d*). Compared with the dark state structure, the C-terminal part of Jα is unstructured and the hydrogen bonds between Gln365 and the carbonyl and amine group of Cys316 are broken, indicating light-induced Jα undocking from the LOV β-sheet and subsequent unfolding as suggested previously (*Herman et al., 2013*) (*Figure 4—figure supplement 1*). Interestingly, the side chains of Cys316 and Leu317 located on strand Gβ also undergo conformational changes upon illumination, which may promote light-induced Jα undocking from the LOV core. Jα seems to contribute directly to the stability of the light state dimer by interacting with A'α of the second protomer. However, the effect of these interactions needs to be interpreted with caution since the conformations of the A′α helices differ between the two protomers (*Figure 4—figure supplement 2*). This could either originate from intrinsic asymmetry of the *Pt*Au1a$_{LOV}$ light-state dimer or from asymmetry induced by crystal contacts (*Figure 4—figure supplement 2*). The formation of the covalent Cys287–FMN adduct and the resulting doming of the isoalloxazine ring at position C4a are supported by the F$_o$-F$_c$ omit map and the 2F$_o$-F$_c$ map (*Figure 4—figure supplement 1*). Light-induced rotamer changes were observed for a variety of residues predominantly in the vicinity of the FMN cofactor (Leu317, Phe331 and Ile333) or on the outer side of strand Iβ (Cys 351) and Gβ (Met 313). Additionally, the highly conserved Tyr266 located on strand Bβ shows a light-induced rotation out of the dimer interface, which is prerequisite for LOV dimerization.

## The LOV and bZIP domains of PtAu1a interact in the dark

To relate light-induced LOV dimerization to the enhanced DNA binding of *Pt*Au1a$_{full}$ in the light and to identify structural elements involved in light signaling, we performed HDX-MS measurements comparing deuterium uptake of dark- and light-adapted *Pt*Au1a$_{full}$ in the absence and presence of DNA as well as of dark- and light adapted *Pt*Au1a$_{LOV}$. Measurements on *Pt*Au1a$_{LOV}$ confirmed the important role of A′α for dimer formation and revealed slightly reduced deuterium incorporation into fast-exchanging amides of A′α in light-adapted *Pt*Au1a$_{LOV}$ (*Figure 4e*). After longer exchange times, all except one of the A′α peptides show the opposite effect and exhibit slightly increased deuterium incorporation. (*Figure 4—figure supplement 3*). Analysis of the peptide overlap showed that the observed destabilization originates from an increased exchange of the amide proton of Phe252 that is located on strand Aβ and is part of the *Pt*Au1a$_{LOV}$ dimerization site that is covered by A′α in the dark state crystal structure (*Figure 4c*). In the dark as well as light state structure, the Phe252 amide proton is in hydrogen bonding distance to the hydroxyl group of Ser268 located between strand Bβ and helix Cα that is also destabilized at later time points, suggesting interaction of these elements. Decreased conformational dynamics is observed for helix Eα encompassing the photoreactive cysteine that forms the covalent Cys287-FMN adduct upon illumination. Structural

stabilization is observed for the end of strand Hβ with its adjacent loop that is in hydrogen bonding distance to Bβ upon dimer formation. Only negligible light-induced differences in deuterium uptake are observed for Jα that was shown to unwind upon illumination (*Herman et al., 2013*) and that is also partially unstructured in the light state crystal structure. This originates from the fact that Jα is highly dynamic already in the dark and amide hydrogen exchange is too rapid to detect a further light-induced increase of the hydrogen exchange rate of Jα due to the limited time resolution of the chosen HDX-MS approach. Similar results were also reported for HDX-MS and nuclear magnetic resonance (NMR) measurements on the isolated LOV2 domain of photoropin 1 from *Avena sativa* (*Winkler et al., 2015*; *Harper et al., 2003*). In addition to Jα, A′α is also highly dynamic and shows a high degree of deuterium incorporation in the dark and light (*Figure 4f*). The moderate protection observed for the crystallographic *Pt*Au1a_LOV light state dimer interface upon illumination is in line with HDX-MS measurements performed on VIVID (*Lee et al., 2014*) and suggests a rapid monomer–dimer interconversion. Thus, the HDX-MS data obtained for *Pt*Au1a_LOV highlight the important role of A′α for light-induced dimerization and support the observed dimer arrangement in the crystal structure also in solution.

In contrast to the weak light-induced changes observed for *Pt*Au1a_LOV, measurements on *Pt*Au1a-_full revealed pronounced destabilization of several elements of the LOV domain upon illumination including Jα and A′α (*Figure 5a–c* and *Figure 5—figure supplement 1*). Additionally, elements within the bZIP DNA-binding domain are significantly destabilized, whereas the N-terminal domain and bZIP–LOV linker region are mainly unaffected by LOV activation. The deuterium exchange behavior determined for LOV domain peptides of light-activated *Pt*Au1a_full and free *Pt*Au1a_LOV are nearly identical, indicating no differences in the conformation or interaction state of the free or effector-coupled LOV domain in the light state (*Figure 5—figure supplement 2*). Therefore, the pronounced differences in deuterium exchange of the LOV and bZIP domains between dark- and light-adapted protein can be explained by a direct interaction between the two domains in the dark, which is broken upon illumination and subsequent LOV dimerization.

Closer inspection of the HDX-MS data suggests the N-terminal part of the leucine zipper as the potential interaction site of the two domains in the dark. Peptides within this region revealed bimodal exchange behavior, indicative of two distinct protein populations with slow interconversion and significantly different deuterium exchange kinetics (EX1 kinetics (*Konermann, Pan, and Liu, 2011*)), which might reflect bZIP–LOV dissociation during labeling (*Figure 5d*). Interestingly, A′α as well as elements of Iβ and the beginning of Jα show a significant increase in deuterium uptake upon illumination and exhibit exchange kinetics similar to those observed for the leucine zipper peptides around residues 176-184, suggesting functional interplay or direct interaction of these elements (*Figure 5b–d*). Structural destabilization is observed for helix Fα and its adjacent linker regions as well as Hβ and the loop between Gβ and Hβ. Since slowly exchanging amides are involved in the exchange of the aforementioned elements, the observed destabilization can be interpreted as an increase in structural dynamics. The N-terminal domain as well as the linker connecting the bZIP and LOV domain show rapid deuterium uptake in the dark and light, which identifies these regions as highly dynamic and probably unstructured as also suggested from secondary structure prediction (*Figure 5—figure supplement 3*). Subtle light-induced structural destabilization was observed for the region between residues 28 and 44 of the N-terminal domain, which suggests that illumination also affects the function of the N-terminal domain.

To study the effect of illumination on the DNA binding affinity of *Pt*Au1a_full and identify regions affected by protein–DNA complex formation, we performed HDX-MS measurements in the presence of DNA. As expected from the EMSA experiments, illumination results in an increased DNA binding affinity of *Pt*Au1a_full, reflected by a pronounced protection of the basic DNA binding region and a decrease in structural dynamics of leucine zipper peptides (*Figure 5e* and *Figure 5—figure supplement 4*). DNA binding and illumination not only affect the bZIP domain but also several elements within the LOV domain, indicating bidirectional allosteric signaling as also reported for the blue light-regulated phosphodiesterase 1 (BlrP1) of *Klebsiella pneumoniae* (*Winkler et al., 2014*). Stabilization of the light-adapted state is observed for A′α, Aβ, Fα, Eα encompassing the photoreactive cysteine and Hβ with its adjacent loop, whereas destabilization is detected for helix Cα, as also observed for the measurements without DNA and in isolated *Pt*Au1a_LOV. Interestingly, the presence of DNA in the dark measurements induces similar effects within the LOV domain as illumination in the absence of DNA, suggesting DNA-induced bZIP–LOV dissociation (*Figure 5—figure*

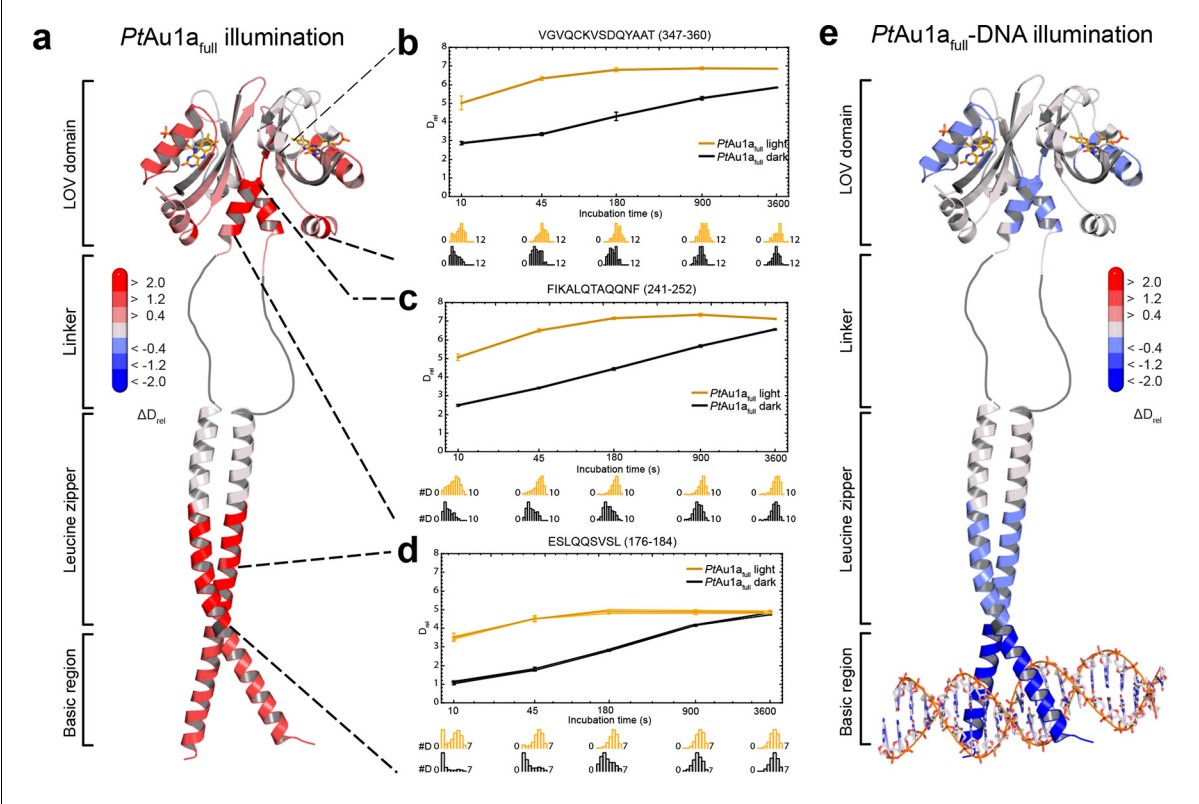

**Figure 5.** HDX-MS data of $PtAu1a_{full}$ in the absence and presence of DNA. (**a**) Changes in deuterium incorporation of $PtAu1a_{full}$ mapped onto the structure of the $PtAu1a_{LOV}$ light state dimer and a model of the bZIP domain. (**b–d**) Deuterium uptake plots of Jα-Iβ, A'α and leucine zipper peptides with $D_{rel}$ plotted against the labeling time for three independent experiments. The estimated abundance distribution of individual deuterated species is shown at the bottom. (**e**) Differences in deuterium incorporation of $PtAu1a_{full}$ in the dark and light state in the presence of DNA mapped onto the $PtAu1a_{LOV}$ light state dimer and a model of the bZIP domain. All evaluated peptides for $PtAu1a_{full}$ in the absence and presence of DNA and their individual deuteration plots are shown in *Figure 5—figure supplement 6* and *Figure 5—figure supplement 7*, respectively. HDX-MS, hydrogen/deuterium-exchange coupled to mass spectrometry; LOV, light-oxygen-voltage.

The following figure supplements are available for figure 5:

**Figure supplement 1.** Effect of illumination on $PtAu1a_{full}$ in the absence of DNA.

**Figure supplement 2.** Comparison of HDX characteristics of LOV domain peptides of $PtAu1a_{full}$ and $PtAu1a_{LOV}$ in the light.

**Figure supplement 3.** Normalized relative deuterium incorporation ($D_{norm}$) of $PtAu1a_{full}$ in the dark.

**Figure supplement 4.** Effect of illumination on $PtAu1a_{full}$ in the presence of DNA.

**Figure supplement 5.** Effect of DNA on LOV domain peptides of $PtAu1a_{full}$ in the dark.

**Figure supplement 6.** Overview of all 80 $PtAu1a_{full}$ peptides evaluated during HDX-MS analysis.

**Figure supplement 7.** Overview of all 80 peptides evaluated during HDX-MS analysis of $PtAu1a_{full}$ in the presence of DNA.

supplement 5). The N-terminal domain of $PtAu1a_{full}$ does not show significant differences in deuterium incorporation in the presence of DNA and seems to play a negligible role for light-dependent DNA binding (*Figure 5—figure supplement 4*).

We obtained an independent confirmation of the suggested bZIP–LOV interaction in the dark from MALS experiments. We incubated $PtAu1a_{LOV}$ together with $PtAu1a_{bZIP}$ upon which the elution volume shifted to a lower volume and the molar mass signal increased slightly compared with

PtAu1a$_{LOV}$ alone (**Figure 6**). Although the detected effects were only subtle and indicate nearly complete complex dissociation during size-exclusion chromatography, they were reproducible in several experiments.

## SAXS measurements reveal light-induced PtAu1a elongation in solution

To analyze light and oligonucleotide induced global conformational changes of PtAu1a$_{full}$ and PtAu1a$_{bZIP-LOV}$ in solution, we performed SAXS measurements. The data obtained for PtAu1a$_{full}$ was of poor quality and did not allow shape reconstructions. Estimation of the radius of gyration (R$_g$) using the Guinier approximation was possible and revealed a light-induced increase of R$_g$ in the measurements without DNA (**Figure 7—source data 1**). Since the SAXS data collected for PtAu1a$_{bZIP-LOV}$ was of high quality and the protein variant encompasses all important structural elements affected by light-activation and DNA binding, we focused on the analysis of the PtAu1a$_{bZIP-LOV}$ data (**Figure 7—figure supplement 1**). As observed for PtAu1a$_{full}$, PtAu1a$_{bZIP-LOV}$ also exhibited a slight increase in the R$_g$ and the maximum particle diameter (D$_{max}$) upon illumination, indicating light-induced protein elongation (**Figure 7—source data 2**).

To obtain information on the overall shape of PtAu1a$_{bZIP-LOV}$ in its light and dark state, we performed *ab initio* modeling using DAMMIN (**Svergun, 1999**). The low-resolution structure reconstructed from the dark measurements matches the length of the bZIP dimer and shows additional density that originates from the two LOV monomers. Rigid body modeling of the PtAu1a$_{bZIP-LOV}$ dark state shows the LOV monomers arranged close to the potential bZIP–LOV interaction site identified by HDX-MS and confirms an interaction of the LOV domain with the leucine zipper in the dark (**Figure 7a**).

Shape reconstructions of PtAu1a$_{bZIP-LOV}$ in the light resulted in a similar but slightly more elongated envelope with less pronounced bulges in the center (data not shown). Since light measurements were performed by short pre-illumination of the protein, a substantial part of PtAu1a$_{bZIP-LOV}$ might not have been light-activated and/or reverted from its light back to its dark conformation during the measurements. Additionally, light-adapted PtAu1a$_{bZIP-LOV}$ exhibits a high degree of flexibility and populates a variety of different conformations. Therefore, it is likely that the obtained envelope represents a mixture of different dark- and light-adapted protein conformations that contribute

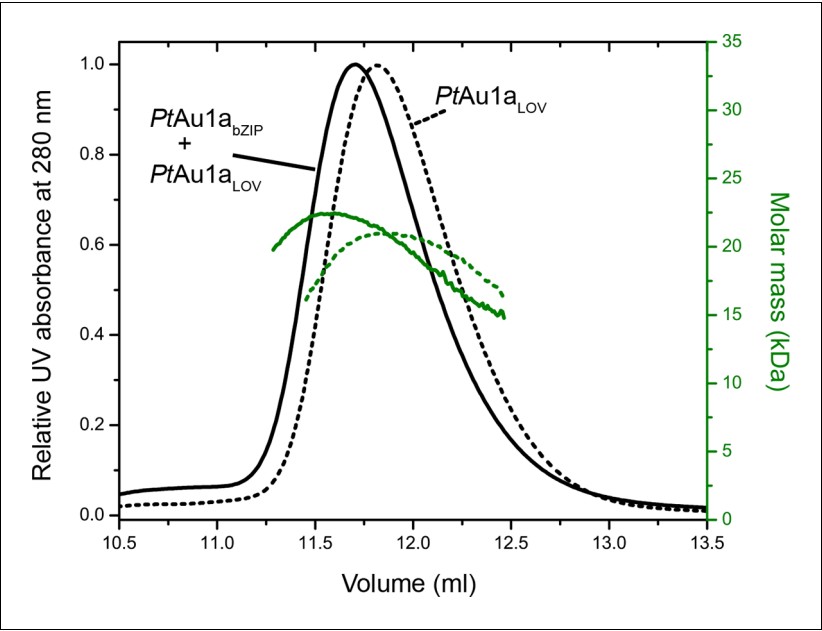

**Figure 6.** Normalized MALS detection of PtAu1a$_{LOV}$ (solid line) alone and PtAu1a$_{LOV}$ together with PtAu1a$_{bZIP}$ (dashed line) fractionated by size-exclusion chromatography in the dark. The MALS-derived molar-mass signals are shown in green. PtAu1a$_{LOV}$ and PtAu1a$_{bZIP}$ interact in the dark, which is reflected by a slight decrease of the elution volume of PtAu1a$_{LOV}$ and an increase of the calculated molar mass signal. LOV, light-oxygen-voltage.

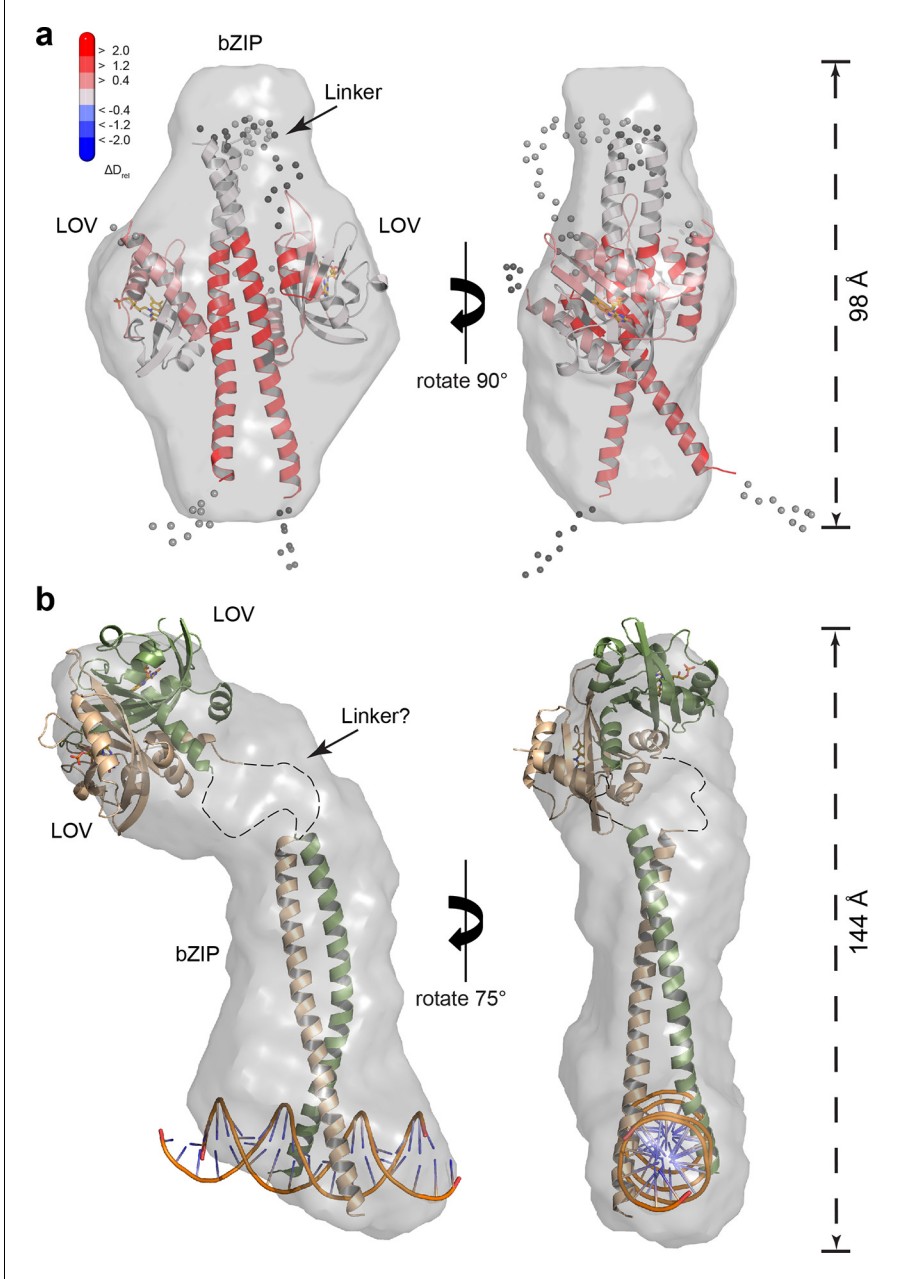

**Figure 7.** SAXS-derived shape reconstructions of $PtAu1a_{bZIP-LOV}$. (**a**) DAMMIN low-resolution envelope calculated for $PtAu1a_{bZIP-LOV}$ in the dark superimposed with an atomic $PtAu1a_{bZIP-LOV}$ dark state model calculated by CORAL. Modelled loops are indicated as dots. The model is colored according to the HDX-MS data obtained for $PtAu1a_{full}$ in the absence of DNA and shows differences in deuterium incorporation in the dark and light state after 10 s of labeling. The results obtained from SAXS-based rigid body modeling, *ab initio* modeling and HDX-MS data agree perfectly and support an interaction between the LOV domain and the leucine zipper of the bZIP domain. (**b**) Shape reconstruction of the $PtAu1a_{bZIP-LOV}$-DNA complex calculated using DAMMIN. The $PtAu1a_{LOV}$ light state dimer and a model of the DNA bound bZIP domain was placed in the envelope by visual inspection. HDX-MS, hydrogen/deuterium-exchange coupled to mass spectrometry; LOV, light-oxygen-voltage; SAXS, small-angle X-ray scattering.

The following source data and figure supplement are available for figure 7:

**Source data 1.** Rg values calculated for $PtAu1a_{full}$ from SAXS data

**Source data 2.** Structural parameters calculated for $PtAu1a_{bZIP-LOV}$ from SAXS data

**Figure supplement 1.** SAXS data for $PtAu1a_{bZIP-LOV}$.

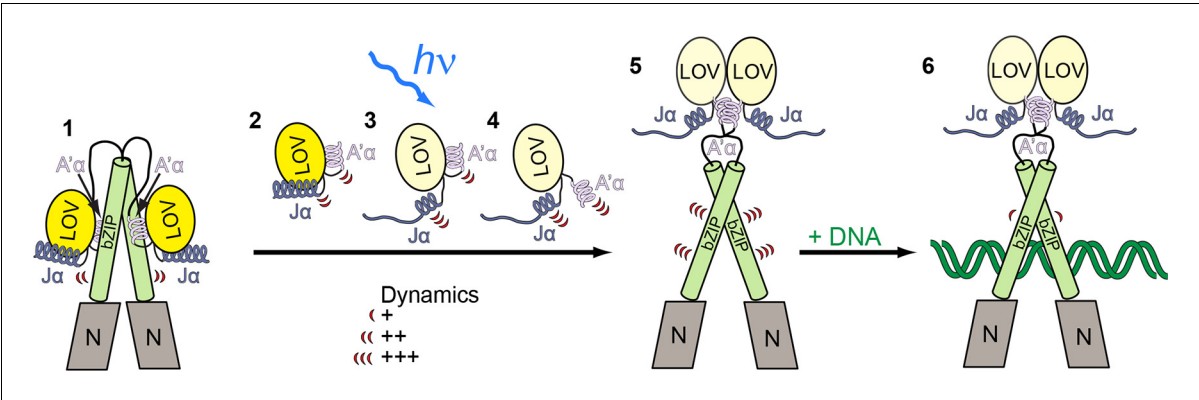

**Figure 8.** Model for light-regulated gene expression by *Pt*Au1a. (**1**) In the dark, *Pt*Au1a is dimeric and the LOV and bZIP domains interact directly thus inhibiting DNA binding of *Pt*Au1a. 2-4: close up of the LOV domain. (**2**) A'α and Jα are attached to the surface of the LOV β-sheet and are highly dynamic even in the dark. (**3**) Illumination with blue light causes Cys287–FMN adduct formation and results in undocking of Jα from the LOV core and increases its structural dynamics. (**4**) Structural changes within the LOV core together with the destabilization of Jα trigger the release of A'α from the dimerization site and also increase its flexibility. (**5**) The LOV domain dissociates from the leucine zipper of the bZIP domain and dimerizes, which results in an increased structural dynamics of the bZIP domain and (**6**) increases the affinity of *Pt*Au1a for its target DNA sequence. The model depicted includes results from FT-IR experiments on *Pt*Au1a-LOV that revealed Jα-dependent A'α release from the LOV core (**Herman and Kottke, 2015**). FT-IR, Fourier transform infrared; FMN, flavin mononucleotide; LOV, light-oxygen-voltage.

simultaneously to the measured SAXS data. For this reason, shape reconstructions of light-activated *Pt*Au1a$_{bZIP-LOV}$ are more difficult to interpret than those of dark-adapted protein. However, light measurements of *Pt*Au1a$_{bZIP-LOV}$ in the presence of DNA clearly show that illumination together with DNA binding induces bZIP–LOV dissociation and results in an elongated protein–DNA complex with $R_g$ and $D_{max}$ values that are significantly larger than those observed for dark-adapted *Pt*Au1a$_{bZIP-LOV}$ (*Figure 7b*).

## Discussion

Aureos are blue light-regulated transcription factors that exhibit an unusual effector–sensor domain topology that is opposite to the domain order found in most LOV proteins and raises the question how signal transmission and light-dependent DNA binding are achieved in these photoreceptors. Our studies on *Pt*Au1a in the presence of its cognate DNA provide structural and functional insights into light-dependent DNA binding of Aureos with implications not only for their biological function, but also for a better understanding of allosteric light-signaling in multidomain LOV proteins.

Crystal structures of the activated light state dimer as well as of the dark state *Pt*Au1a$_{LOV}$ monomer reveal the molecular mechanism of blue light-dependent Aureo LOV dimerization. The N- and C-terminal A'α and Jα helices flanking the LOV core play a central role in this process and are directly affected by illumination as also observed for phototropin-LOV2 from *Av. sativa* and *A. thaliana* (*Zayner, Antoniou, and Sosnick, 2012*; *Takeda et al., 2013*; *Harper, Neil, and Gardner, 2003*). In the dark, A'α covers the hydrophobic dimerization site of the LOV β-sheet that becomes exposed upon illumination and thus enables LOV dimerization. These results provide a molecular mechanism for recent size-exclusion chromatography experiments performed on isolated *Pt*Au1a LOV constructs lacking A'α and/or Jα that also imply that A'α covers the LOV dimerization site in the dark (*Herman and Kottke, 2015*; *Herman et al., 2013*). However, not only A'α, but also Jα is affected by blue light illumination. It undocks from the LOV β-sheet and partially unfolds. The light-induced structural changes observed for A'α and Jα are in line with Fourier transform infrared (FT-IR) spectroscopy studies that also indicate light-induced structural changes in both helices (*Herman and Kottke, 2015*; *Herman et al., 2013*). Interestingly, it was reported that A'α is only affected by illumination in the presence and after unfolding of Jα, whereas Jα unfolds even in the absence of A'α (*Herman and Kottke, 2015*). Since a direct interaction between the two helices is not observed in the dark state *Pt*Au1a$_{LOV}$ crystal structure, the interdependence between A'α and Jα indicates an allosteric interplay between the two helices as suggested previously (*Herman and Kottke, 2015*).

**Table 1.** Data collection and refinement statistics.

| | *Pt*Au1a$_{LOV}$ dark | *Pt*Au1a$_{LOV}$ light |
|---|---|---|
| Data collection | | |
| Space group | $P2_12_12_1$ | $P3_221$ |
| Cell dimensions | | |
| *a, b, c* (Å) | 64.4, 69.2, 74.6 | 108.6 108.6 104.7 |
| α, β, γ (°) | 90.0, 90.0, 90.0 | 90.0 90.0 120.0 |
| Resolution (Å) | 50–2.5 (2.5–2.6) * | 50–2.7 (2.7–3.2) |
| $R_{meas}$ | 13.2 (55.1) | 10.0 (63.7) |
| *I* / σ*I* | 15.57 (4.50) | 13.01 (2.46) |
| Completeness (%) | 99.9 (100) | 98.9 (98.1) |
| Redundancy | 13.1 (13.5) | 5.0 (5.0) |
| Refinement | | |
| Resolution (Å) | 48.7-2.5 | 45.7-2.7 |
| No. reflections | 12,016 | 19,785 |
| $R_{work}$ / $R_{free}$ | 0.196/0.254 | 0.209/0.243 |
| No. atoms | 2237 | 2164 |
| Protein | 2108 | 2100 |
| Ligand/ion | 74 | 62 |
| Water | 55 | 2 |
| *B* factors | 33.3 | 74.7 |
| Protein | 33.4 | 74.9 |
| Ligand/ion | 32.6 | 69.1 |
| Water | 32.2 | 72.3 |
| r.m.s. deviations | | |
| Bond lengths (Å) | 0.007 | 0.009 |
| Bond angles (°) | 1.109 | 1.090 |

*Values in parentheses are for highest-resolution shell. One crystal was used for data measurements of *Pt*Au1a$_{LOV}$ in its dark and light-adapted state, respectively.

Allosteric signaling occurs via the LOV β-sheet and involves the central strand Iβ, which is covalently coupled to Jα and also interacts with A'α in the dark. The important role of the LOV β-sheet in light-signaling is supported by the fact that several β-sheet residues show light-induced rotamer changes. Since the dimer arrangement observed in the *Pt*Au1a$_{LOV}$ light state structure differs from the dimeric assembly observed in the recently determined dark state crystal structure of *Vf*Au1 LOV (*Mitra et al., 2012*), it can be concluded that a combination of light-induced structural rearrangements within the LOV core together with undocking of Jα from the β-sheet are required to trigger the release of A'α from the LOV core, which ultimately induces the formation of the biologically relevant light state dimer. In contrast to the consecutive structural changes reported for the flanking helices of *Pt*Au1a LOV, A'α and Jα of phototropin-LOV2 exhibit structural changes independently of each other (*Zayner et al., 2012*; *Takeda et al., 2013*), which implies differences in the signaling mechanism of Aureos and Phototropins.

Two models for light-dependent DNA binding of *Vf*Au1 have been proposed in the literature where LOV dimerization plays a central role. According to the first model, *Vf*Au1 is dimeric regardless of the light conditions and additional light-induced LOV dimerization enhances its affinity for DNA (*Toyooka et al., 2011*). In the second model, DNA binding not only depends on the light, but also on the redox conditions (*Hisatomi et al., 2014*). Under reducing conditions, *Vf*Au1 exists as a monomer and blue light illumination changes the oligomerization state from monomers to dimers, thus increasing the affinity for its target DNA sequence. Under oxidizing conditions, intermolecular

disulfide bonds are formed between the bZIP domains and the bZIP–LOV linker regions, which induce *Vf*Au1 dimerization and enable light-independent DNA binding. We point out that *Pt*Au1a does not possess cysteine residues outside the LOV domain and is therefore unable to form such disulfide bonds. Consequently, an influence of the redox conditions on the DNA binding of *Pt*Au1a can be ruled out. Concerning light-induced changes in the oligomerization state, we could not detect indications for light-induced *Pt*Au1a$_{full}$ and *Pt*Au1a$_{bZIP-LOV}$ oligomerization in any of our experiments. Instead, our data suggests a mechanism where the LOV domain directly interacts with the leucine zipper region of the bZIP domain in the dark, impeding its DNA binding function. Our HDX-MS data suggest that the LOV–bZIP interaction might occur via the LOV β-sheet and directly involves A'α as well as Jα. Upon blue light illumination, structural changes occur within the LOV core that are transmitted to the flanking helices and result in bZIP–LOV dissociation and subsequent LOV dimerization, thus increasing the affinity of *Pt*Au1a for its target DNA sequence (*Figure 8*). Such an allosteric signaling mechanism not only explains the absence of changes in structural dynamics of the bZIP–LOV linker region upon illumination, but also explains the need for the > 30 amino acids long linker in Aureos, which is required to enable direct bZIP–LOV interaction. Our model is further supported by recent data on N-terminally truncated and modified *Vf*Au1 constructs that also indicate an interaction between the leucine zipper and the LOV domain in the dark, although it was hypothesized that this interaction stabilizes the monomeric form of the synthetic proteins (*Nakatani and Hisatomi, 2015*). Moreover, a mechanism involving light-induced release of the LOV domain from its interaction site on the bZIP domain is consistent with the general concept of LOV and PAS domain signaling via the β-sheet surface (*Harper et al., 2003*; *Zoltowski et al., 2007*; *Möglich et al., 2009*; *Nash et al., 2011*; *Rivera-Cancel et al., 2014*). Since the cellular concentration of *Pt*Au1a is not known, we cannot completely exclude a regulatory mechanism where light-induced LOV dimerization additionally influences the oligomerization state and thus DNA binding of *Pt*Au1a *in vivo*.

So far, the very limited number of light state and multi-domain LOV protein crystal structures has hampered our understanding of allosteric light signaling in this class of photoreceptors. Detailed structural and functional characterization of the light- and dark-adapted states are required for elucidating how the light signal is transmitted from the light-sensing LOV to the effector domain. Our data for *Pt*Au1a not only contribute to the understanding of the modularity and allosteric light signaling in LOV proteins, but also provides important information for rational and structure-guided design of new optogenetic tools. In fact, our data also explain the underlying working mechanism of recently engineered Aureo LOV containing light-activatable synthetic transcription factors and receptor tyrosine kinases that are based on light-dependent homo-dimerization (*Yang et al., 2013*; *Grusch et al., 2014*). These successful applications of Aureo LOVs highlight their efficacy as photodimerizers. However, the full potential of Aureo LOV domains for the design of optogenetic devices has not been exploited yet. In addition to their dimerization ability, the concerted undocking of A'α, Jα as well as of the bZIP domain from the LOV core can be used for the design of new optogenetic tools. The light-sensitive bZIP–LOV interaction observed in Aureos allows combining the engineering strategies developed for leucine zippers and coiled coils with the light-sensing function of LOV domains. This unique combination of different functionalities provided by Aureos offers new design strategies for robust and tightly controllable molecular switches that allow precise spatial and temporal control of biological processes.

## Material and methods

### Cloning of the full-length and truncated *Pt*Au1a variants

The pETM-11 plasmids encoding *Escherichia coli* codon optimized *Pt*Au1a$_{full}$ (Epoch Life Science) and *Pt*Au1a$_{LOV}$ were kindly provided by H. Janovjak, IST Austria. We generated *Pt*Au1a constructs representing *Pt*Au1a$_{bZIP-LOV}$ and *Pt*Au1a$_{bZIP}$ by polymerase chain reaction (PCR) amplification using pETM-11 *Pt*Au1a$_{full}$ as template and the following primer pairs: 148_fw (5'-ATATCCATGGGAATGT-CTGAGCAGCAGAAAGTGG-3') and 378_rv (5'-ATATGCGGCCGCTTAGTCTTCATCGT-CATTGGCTG-3') for *Pt*Au1a$_{bZIP-LOV}$ and 148_fw and 212_rv (5'-ATATGCGGCCGCTTAAGCGGAATCGATCAGGGTG-3') for *Pt*Au1a$_{bZIP}$. The PCR products were cloned into the pETM-11 vector using *Nco*I and *Not*I restriction sites. The resulting *Pt*Au1a$_{bZIP-LOV}$

and $PtAu1a_{bZIP}$ as well as the $PtAu1a_{full}$ and $PtAu1a_{LOV}$ constructs carry an N-terminal hexahistidine tag followed by a *Tobacco Etch Virus* (TEV) protease cleavage site.

## Protein expression and purification

Chemically competent *E. coli* BL21 (DE3) cells (Invitrogen) were transformed with the respective *Pt*Au1a plasmid DNA. Protein expression was induced with 0.2 mM isopropyl $\beta$-D-1-thiogalactopyranoside at an optical density of 0.8. All protein constructs were expressed overnight at 18°C in the dark in Lysogeny broth medium supplemented with 30 µg ml$^{-1}$ kanamycin. Cells were harvested by centrifugation and the cell pellets were resuspended in buffer A (20 mM 4-(2-hydroxyethyl)-1-piperazineethanesulfonic acid (HEPES) pH 7.5, 300 mM NaCl, 40 mM imidazole, 5% (w/v) glycerol) including cOmplete Protease-Inhibitor Cocktail (Roche). The cells were lysed using a microfluidizer (Microfluidics) and the lysates were clarified by ultracentrifugation at 185,500 g at 4°C for 1 hr. The supernatant was loaded onto an Ni$^{2+}$–NTA Superflow (Qiagen) affinity column pre-equilibrated with buffer A. The resin was washed with 10 column volumes (CV) of buffer A and the bound proteins were eluted using 5 CV of buffer A supplemented with 160 mM imidazole. The protein containing fractions of $PtAu1a_{full}$, $PtAu1a_{bZIP-LOV}$ and $PtAu1a_{LOV}$ were dialyzed at 4°C overnight against 1 L buffer B (20 mM HEPES pH 7.5, 50 mM NaCl, 2 mM dithioerythritol, 2 mM ethylenediaminetetraacetic acid (EDTA), 5% (w/v) glycerol) and in parallel the hexahistidine tag was removed from the protein using TEV protease (1:30 molar ratio of TEV:protein). The cleaved tag and the histidine-tagged TEV protease were removed from the *Pt*Au1a solutions by rechromatography on Ni$^{2+}$–NTA resin and the flow through was used for further purification. After Ni$^{2+}$–NTA chromatography, $PtAu1a_{bZIP}$ was directly loaded on a HiTrap heparin column (GE Healthcare) using buffer C (20 mM HEPES pH 7.5, 100 mM NaCl, 10 mM MgCl$_2$, 5% (w/v) glycerol) as running buffer and eluted in a gradient to 100% buffer C supplemented with 950 mM NaCl. $PtAu1a_{full}$ and $PtAu1a_{bZIP-LOV}$ were loaded onto a MonoS column (GE Healthcare) using buffer D (50 mM 2-(N-morpholino)ethansulfonic acid (MES) pH 6.0, 50 mM NaCl) as running buffer and eluted in a gradient to 100% buffer D supplemented with 950 mM NaCl. All *Pt*Au1a variants were concentrated using centrifugal filter units (Amicon, Millipore) and the LOV domain containing variants were reconstituted with FMN (Sigma-Aldrich) by incubation with a five-fold excess of FMN for 1 h at 4°C in the dark. Subsequently, $PtAu1a_{full}$, $PtAu1a_{bZIP-LOV}$ and $PtAu1a_{LOV}$ were subjected to gel filtration on a Superdex 200 ($PtAu1a_{full}$) or Superdex 75 ($PtAu1a_{bZIP-LOV}$ and $PtAu1a_{LOV}$) (GE Healthcare) column equilibrated in buffer C .

For SAXS analysis, $PtAu1a_{full}$ and $PtAu1a_{bZIP-LOV}$ as well as their DNA complexes were buffer exchanged on a Superdex 200 Increase 10/300 GL column (GE Healthcare) equilibrated in buffer C. To efficiently form the $(PtAu1a_{full})_2$-DNA and $(PtAu1a_{bZIP-LOV})_2$-DNA complex the proteins were incubated for 5 min under blue light illumination together with a 1.5-fold molar excess of 21-bp DNA and subsequently subjected to gel-filtration. The peak fractions of the eluting protein–DNA complexes were pooled and incubated at 4°C overnight to allow back conversion to the dark state.

## Oligonucleotide purification

24 bp blunt end DNA probes encompassing a TGACGT bZIP binding motif (5'- TGTAGCGT-CTGACGTGGTTCCCAC-3', the binding motif is underlined) of the *P. triccornutum dsCYC2* promoter region or a random DNA sequence (5'-AGTGGGTCATTGCAAGTAGTCGAT-3') as well as a 21 bp DNA probe with single base-pair overhangs encompassing the binding sequence (fw: 5'-ATAGCGTCTGACGTGGTTCCC-3', rv: 5'-TGGGAACCACGTCAGACGCTA-3') were ordered as single strands and resuspended in annealing buffer containing 10 mM Tris pH 7.0, 500 mM NaCl, 2.5 mM MgCl$_2$, 1 mM EDTA. Complementary DNA strands were annealed together in equimolar amounts by heating to 95°C and gradual cooling to room temperature overnight. For purification, the annealed DNA probes were diluted in buffer E (20 mM HEPES pH 7.5, 50 mM NaCl, 5% (w/v) glycerol) and loaded on a MonoQ column (GE Healthcare) equilibrated in buffer E. The DNA was eluted from the column driving a gradient to 100% buffer E supplemented with 950 mM NaCl. Subsequently, the DNA probes were subjected to size-exclusion chromatography on a Superose 6 (GE Healtcare) column equilibrated in buffer C. The DNA probes were concentrated by ultrafiltration using a 10-kDa cut off centrifugal filter unit (Amicon Ultra-4, Millipore).

## Multi-angle light-scattering experiments

Individual LOV domain containing *Pt*Au1a variants were pre-incubated at 20°C in the dark or under continuous blue light illumination (400 μW cm$^{-2}$ at 450 nm) from a royal blue (455 nm) collimated LED lamp (Thorlabs) for 20 min. 100 μl of a 150 μM protein solution was subjected to size-exclusion chromatography at RT on a Superdex 200 10/300 GL column (GE Healthcare) equilibrated in buffer C. For dark and light experiments, the column was kept in the dark or continuously illuminated with blue light during the gel-filtration runs. To investigate the interaction between *Pt*Au1a$_{LOV}$ and *Pt*Au1a$_{bZIP}$, the proteins were mixed in a molar ratio of 1:1 and concentrated using centrifugal filter units (Amicon, 3 kDa cut off). 100 μl of a 75 μM complex solution (based on a theoretical (*Pt*Au1a$_{LOV}$-*Pt*Au1a$_{bZIP}$)$_2$ complex) was subjected to size-exclusion chromatography at RT on a Superdex 200 10/300 GL column (GE Healthcare) equilibrated in buffer C. The high performance liquid chromatography (HPLC) (Waters) setup was connected to a MALS detector (Dawn Heleos, Wyatt Technology) combined with a refractive-index detector (Waters). Data analysis was performed using the *ASTRA* software (Wyatt Technology), providing estimates for the molar mass of the different *Pt*Au1a variants and the (*Pt*Au1a$_{LOV}$-*Pt*Au1a$_{bZIP}$)$_2$ complex.

## Electrophoretic mobility shift assays

24 bp DNA probes (50 nM) were incubated in buffer C supplemented with 5% (w/v) glycerol and varying amounts of purified *Pt*Au1a$_{full}$ in a total volume of 5 μl. The protein DNA mixtures were incubated at RT for 20 min in the dark or under continuous blue light illumination (400 μW cm$^{-2}$ at 450 nm) and then separated on 10% Tris-glycine-EDTA (TGE) gels, pH 9.0, supplemented with 10 mM MgCl$_2$ using buffer containing 25 mM Tris pH 9.0, 190 mM glycine, 10 mM MgCl$_2$ and 1 mM EDTA. Gel runs were performed at 4°C for 90 min (15 V cm$^{-1}$) in the dark or under continuous blue light illumination (30 μW cm$^{-2}$). Gels were stained with GelRed (Biotinum) for DNA visualization. To exclusively visualize DNA and cancel out flavin fluorescence in the EMSA experiments, the gels were imaged using green excitation light and a 605/50 nm emission filter (ChemiDoc MP, Biorad).

## Protein crystallization and structure elucidation

Crystallization of *Pt*Au1a$_{LOV}$ was performed at 20°C. Dark state crystals were grown in sitting-drop geometry with unreconstituted *Pt*Au1a$_{LOV}$ protein (chromophore occupancy ~55%, protein buffer: 20 mM Tris pH 8.0, 100 mM NaCl, 10% (w/v) glycerol) using a protein concentration of 16 mg ml$^{-1}$ and 40% (v/v) ethylene glycol and 0.1 M sodium acetate pH 4.5 as reservoir solution. Orthorhombic crystals appeared after 1 day and were harvested after 5 days. Dark state crystals were flash-cooled in liquid nitrogen under safe-light conditions without further cryoprotection.

*Pt*Au1a$_{LOV}$ light state crystals were grown with FMN-reconstituted protein overnight in hanging-drop vapour-diffusion geometry under continuous blue light illumination (50 μW cm$^{-2}$ at 450 nm). Trigonal crystals started to grow from a mixture of 0.66 μl protein solution (12 mg ml$^{-1}$), 0.66 μl reservoir solution consisting of 3.5 M sodium formate pH 7.0 and 0.66 μl 0.1 M hexamine cobalt (III) chloride. Prior to flash-cooling in liquid nitrogen, crystals were incubated in reservoir solution supplemented with 20% (v/v) glycerol.

Diffraction data of *Pt*Au1a$_{LOV}$ dark and light state crystals were collected on beamline P11 at PETRA III, DESY (Hamburg, Germany) and beamline X10SA at the Swiss Light Source (Villigen, Switzerland) at 100 K and wavelengths of 0.9780 and 0.9795 Å, respectively. Data were processed using the *XDS* program suite (*Kabsch, 2010*). To minimize X-ray-induced reduction of the covalent adduct formed in the light-adapted protein, *Pt*Au1a$_{LOV}$ light state data was collected from three different spots on a single crystal and merged to obtain a complete data set. *Pt*Au1a$_{LOV}$ dark and light state data were phased by molecular replacement using *Phaser* (*McCoy et al., 2007*) in *CCP4* and *Phaser* in *PHENIX*, respectively, with the Aureochrome 1 LOV domain (residues 219-317) from *Vaucheria frigida* (PDB code 3UE6, molecule A) and the *Pt*Au1a$_{LOV}$ dark state structure without A′α helix (residues 251-372) as the search models. The missing segments flanking the LOV core were manually built in *Coot* (*Emsley et al., 2010*) and the structure refined in cycles of *phenix.refine* (*Afonine et al., 2012*) refinement and manual re-building. R$_{free}$ values for the *Pt*Au1a$_{LOV}$ dark and light state data sets were computed from 8% and 5% randomly chosen reflections not used during the refinement, respectively. The topology and parameter files for the covalent Cys287-FMN adduct were obtained by quantum chemical calculations (*Fedorov et al., 2003*). Model quality was analyzed

using the *MolProbity* (*Emsley et al., 2010*) validation tool as implemented in *PHENIX*. Both final models contained no outliers in the Ramachandran plot, with 100% of the residues in the favored region for the dark state structure of *Pt*Au1a$_{LOV}$ and 96.2% for the light state structure. Atomic coordinates of the structures and structure-factor amplitudes have been deposited in the Protein Data Bank as entries 5DKK and 5DKL. Analysis of the *Pt*Au1a$_{LOV}$ light state dimer interface and the BSAs was performed using the Protein Interfaces, Surfaces and Assemblie service (PISA; v1.51) (*Krissinel and Henrick, 2007*).

## Hydrogen-deuterium exchange mass spectrometry

Aliquots of 2 µl at final concentrations of 250 µM *Pt*Au1a$_{full}$, 250 µM *Pt*Au1a$_{full}$ together with a 1.2-fold excess of 24 bp DNA on the basis of a (*Pt*Au1a$_{full}$)$_2$-DNA stoichometry and 250 µM *Pt*Au1a$_{LOV}$ were pre-incubated at 20°C for 60 s in the dark or under continuous blue light illumination (400 µW cm$^{-2}$ at 450 nm) in buffer D. The corresponding light conditions were maintained throughout the labeling reactions, which were prepared in triplicates for all experiments. Hydrogen-deuterium exchange was initiated by 1:20 dilution of the samples in buffer D prepared with D$_2$O and glycer(ol-d$_3$) pD 7.5 at 20°C. Aliquots of 6 µl were removed after 10 and 45 s, and 3 , 15 and 60 min and the labeling reaction terminated by quenching with 56 µl ice cold buffer containing 200 mM ammonium formic acid pH 2.6 and 2.8 M urea. Deuterated samples were injected into a cooled HPLC setup and digested on an immobilized pepsin column (Poroszyme, Life Technologies) kept at 10°C. All subsequent steps were carried out in a water bath at 0.5 ± 0.1°C. The generated peptides were desalted on a 2 cm C18 guard column (Discovery Bio C18, Sigma) and separated during a 7 min acetonitrile gradient (15–50%) in the presence of 0.6% (v/v) formic acid on a reversed phase column (XR ODS 75 × 3 mm, 2.2 µM; Shimadzu). Eluting peptides were infused into a maXis electrospray ionization-ultra high resolution-time-of-flight mass spectrometer (Bruker) and deuterium incorporation was analyzed and quantified using the Hexicon 2 software package (*Lindner et al., 2014*).

## Small angle X-ray scattering

SAXS measurements were performed at the X12SA cSAXS beamline at the Swiss Light Source (Villigen, Switzerland). Measurements were performed in buffer C at protein concentrations of 10, 5, and 2.5 mg/ml for *Pt*Au1a$_{full}$; 9, 5, and 2.5 mg/ml for *Pt*Au1a$_{bZIP-LOV}$; 5.8 and 2.9 mg/ml for the (*Pt*Au1a$_{full}$)$_2$-DNA complex and 6.2 and 3.1 mg/ml for the (*Pt*Au1a$_{bZIP-LOV}$)$_2$-DNA complex. Samples were filled and mounted in Ø 1-mm-quartz capillaries in the dark and kept at 10°C throughout the experiments. For light activation of the LOV domain, the samples were briefly pre-illuminated with a blue LED ($\lambda_{max}$ = 455 nm; Thorlabs). Data acquisition using 11.2 keV photons was performed in 500 µm steps along the capillary with 10 × 0.5 s exposure at 20 different positions. Scattered X-rays were recorded with a Pilatus 2M detector. Data was collected from the buffer alone and subsequently from the protein and protein–DNA complexes from the identical position on the same capillary. Measurements were performed in the dark with and without pre-illumination of the samples. The scattering vector q is defined as q = $4\pi$ sin($\theta$) $\lambda^{-1}$.

For data analysis, all diffraction images were azimuthally integrated, averaged and the buffer signal was subtracted from that of the buffered protein solution. For DAMMIN *ab initio* and CORAL (*Petoukhov et al., 2012*) rigid body modeling *Pt*Au1a$_{bZIP-LOV}$ dark state data from the 5 and 9 mg/ml measurements was merged at q = 0.13 A$^{-1}$ and (*Pt*Au1a$_{bZIP-LOV}$)$_2$-DNA complex light data was merged at q = 0.10 A$^{-1}$. Distance distribution functions p(r) and maximum particle diameters D$_{max}$ were calculated using the program GNOM (*Svergun, 1992*).

Shape reconstructions were performed using the *ab initio* bead-modelling program DAMMIN (*Svergun, 1999*). Ten independent models generated for the dark state of *Pt*Au1a$_{bZIP-LOV}$ and the (*Pt*Au1a$_{bZIP-LOV}$)$_2$-DNA light complex without enforcing symmetry (P1) were superimposed and averaged using the programs DAMSUP (*Volkov and Svergun, 2003*) and DAMAVER (*Volkov and Svergun, 2003*). Finally, the averaged shapes were filtered using the program DAMFILT (*Volkov and Svergun, 2003*).

Molecular modelling was performed using the program CORAL (*Petoukhov et al., 2012*) and the high resolution structure of *Pt*Au1a$_{LOV}$ in its dark state and a homology model of the bZIP domain on the basis of the c-Fos-c-Jun heterodimer (PDB: 1FOS). Modelling was applied keeping the bZIP

dimer fixed and allowing free positioning of the two LOV monomers as well as building of the missing parts.

## Microscale thermophoresis

Dimerization of $PtAu1a_{LOV}$ in the dark was quantified by microscale thermophoresis using a Monolith NT.115 (Nanotemper). The protein was randomly labeled at the amine positions using the NHS-reactive red fluorescent dye DY-647 (MoBiTec) according to the labeling protocol of Nanotemper. A 1:2 dilution series of unlabeled $PtAu1a_{LOV}$ was prepared over an appropriate concentration range using a buffer containing 20 mM HEPES pH 7.5, 100 mM NaCl and 10% (w/v) glycerol and were mixed with equivalent volumes of labeled $PtAu1a_{LOV}$ (final concentration 20 nM). Measurements were performed in the dark using standard treated capillaries. Data of three individual experiments were averaged and evaluated using the quadratic equation of the law of mass action with the constraint of a fixed labelled species concentration.

## UV/VIS spectroscopy

Dark state recovery kinetics of the different LOV domain containing $PtAu1a$ variants were measured at 25°C in buffer D using a Varioskan Flash multimode reader (Thermo Scientific). Samples were pre-illuminated with a blue LED ($\lambda_{max}$ = 455 nm; Thorlabs) for 3 min and subsequently the absorbance was measured at 445 nm. Measurements for all LOV domain containing $PtAu1a$ variants were performed at a protein concentration of 20 µM. Data of three independent measurements were averaged and evaluated by fitting exponential functions.

## Acknowledgements

We thank the Heidelberg team for data collection at the beamlines X10SA and X12SA (cSAXS) at the Swiss Light Source (Villigen, Switzerland) as well as at the beamline P11 at PETRA III, DESY (Hamburg, Germany). We are grateful to A. Meinhart for help with X-ray data analysis and R. Lindner and A. Winkler for help with HDX-MS experiments and data analysis. We are grateful to A. Menzel (SLS, Switzerland) for help with SAXS measurements and T. Domratcheva for performing quantum chemical calculations that allowed generation of the parameter and topology files used in the refinement of the $PtAu1a_{LOV}$ light state dimer structure. We thank R. Lindner for discussion and comments on the manuscript and acknowledge C. Roome for IT support and H. Janovjak (IST Austria) for providing the pETM-11 $PtAu1a_{full}$ and pETM-11 $PtAu1a_{LOV}$ plasmids. We acknowledge financial support by the Max Planck Society, the Hartmut Hoffmann-Berling International Graduate School of Molecular and Cellular Biology (UH) and the German Research Foundation (DFG): FOR526 to IS.

## Additional information

### Funding

| Funder | Grant reference number | Author |
|---|---|---|
| Deutsche Forschungsgemeinschaft | FOR526 | Ilme Schlichting |
| Max-Planck-Gesellschaft | | Ilme Schlichting |
| Hartmut Hoffmann-Berling International Graduate School of Molecular and Cellular Biology | PhD Fellowship | Udo Heintz |

The funders had no role in study design, data collection and interpretation, or the decision to submit the work for publication.

### Author contributions

UH, designed the project, performed and analyzed the experiments, solved the $PtAu1a_{LOV}$ crystal structures and wrote the manuscript; IS, supported design of the project and gave advice during preparation of the manuscript

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
