## [Decision Letter]

Thank you for submitting your work entitled "Blue light-induced LOV domain dimerization enhances the affinity of Aureochrome1a for its target DNA sequence" for consideration by *eLife*. Your article has been reviewed by John Kuriyan (Senior editor) and three reviewers, one of whom, Volker Dötsch is a member of our Board of Reviewing Editors. The following peer reviewer has agreed to reveal his identity: Keith Moffat.

The reviewers have discussed the reviews with one another and the Reviewing editor has drafted this decision to help you prepare a revised submission.

Summary:

LOV proteins are ubiquitous sensory transduction elements that are of widespread relevance to pathogenicity, agricultural production and optogenetic tool design. Further, shared mechanisms with PAS domain containing proteins have positioned LOV proteins as a model of allosteric signal transduction in diverse organisms. Currently, structural information on dark- and light-states of LOV proteins is unavailable for multi-domain containing LOV photoreceptors. For these reasons, concrete evidence of allosteric mechanisms is currently unavailable.

The paper by Heintz and Schlichting describes the structural characterization of the LOV transcription factor Aureochrome1. In contrast to other LOV domain-containing transcription factors, this particular one shows a different organization of its domains. The authors use size-exclusion chromatography, deuterium exchange MS measurements, SAXS measurements and x-ray crystallography to build a model of how illumination with blue light enhances the DNA binding affinity. According to their model, the LOV domain in the dark state binds to the bZIP DNA-binding domain. In the light state soem structural elements of the LOV domain are partially unfolded, allowing for a structural rearrangement that releases the LOV domain and triggers its dimerization.

This is an elegant and interesting model of a complex transcription factor. In the absence of a high resolution structure the authors use a combination of different biophysical methods to characterize the structural changes. All reviewers agreed that the results are highly relevant and interesting and that the paper is clearly written.

Some questions, however, remain that should be addressed in a revised manuscript.

Essential revisions:

1) In Figure 1, the differences between the dark and the light state on the gel filtratoin column are very tiny. If illumination triggers the dimerization of the LOV domain, this should shift the equilibrium towards the dimer. The kD of LOV domain dimerization is rather weak (13 µM), however, the local high concentration when in the bZIP dimer should be high and a significant stabilization of the complex should be expected. Is there an explanation for this very weak effect? In addition, the void volume of the column should be indicated in the figures.

2) In the subsection “Blue light illumination induces dimerization of *Pt*Au1aLOV, but not of *Pt*Au1a_full_ and *Pt*Au1a_bZIP-LOV_”. The authors note a smaller than expected apparent MW for the *Pt*Au1a dimers. Such data is consistent with their H-D exchange data indicating a fast exchanging dimer. Indeed studies of the isolated LOV domain of *Pt*Au1a demonstrated fast exchange characteristics by SEC (Herman et al. Biochemistry 2015), where the elution volume and apparent MW were concentration dependent. Similar data was previously shown for *N. crassa* VVD (Zoltowski et al., Biochemistry 2008) and appears to be common in LOV proteins.

Did the authors examine the concentration dependence of the apparent MW by SEC? Alternatively the lower MW than expected could be due to the relatively fast photocycle (15-25 minutes).

3) Figure 7: Why is there a kink in the SAXS density? According to the model this should be a symmetric dimer with a flexible linker.

---

## [Author Response]

*Essential revisions:*

*1) In Figure 1, the differences between the dark and the light state on the gel filtratoin column are very tiny. If illumination triggers the dimerization of the LOV domain, this should shift the equilibrium towards the dimer. The kD of LOV domain dimerization is rather weak (13 µM), however, the local high concentration when in the bZIP dimer should be high and a significant stabilization of the complex should be expected. Is there an explanation for this very weak effect? In addition, the void volume of the column should be indicated in the figures.*

We thank the reviewers for pointing out this important aspect and think that the reviewers refer to Figure 2 instead of Figure 1. The observed differences between the dark and light state of the bZIP domain containing *Pt*Au1a variants on the gel-filtration column are indeed very small and the respective dimers dissociate rapidly in the dark and light. We initially also expected a stabilization of the complex upon light-activation and subsequent LOV dimerization. However, we could not observe such stabilization for the *Pt*Au1a_full_ and *Pt*Au1a_bZIP-LOV_ dimers in any of our SEC-MALS experiments. We did not observe light-induced dimerization of *Pt*Au1a either, which was suggested previously for *Vf*Au1 (Hisatomi et al. 2014). Our SEC-MALS data indicate that the *Pt*Au1a bZIP domain is a weak dimerization module in the absence of DNA. Indeed, it is even possible that the basic DNA binding region negatively affects bZIP dimerization due to strong repulsive electrostatic interaction between the positively charged subunits. The bZIP containing *Pt*Au1a variants show a low kinetic stability and mostly elute as monomers from the gel- filtration column. Since the majority of molecules elutes as monomers, a high local LOV domain concentration, which might contribute to dimer stability, is only present in a small fraction of the molecules. Additionally, the photocycle of the bZIP domain containing *Pt*Au1a constructs is relatively “fast” and a significant part of the molecules decays from the adduct- back to the dark-state during gel filtration. Direct comparison of the experiments performed for *Pt*Au1a_LOV_ and the bZIP domain containing *Pt*Au1a variants is complicated by the fact that the presence of the bZIP domain accelerates the recovery kinetic of the LOV domain about 1.8-fold (see Figure 1). Interpretation of the SEC elution profiles and potential shifts to lower and larger retention volumes of *Pt*Au1a is hardly possible, because of the intrinsic flexibility and large unstructured regions that enable *Pt*Au1a to adopt multiple conformations. We want to emphasize that interpretation of shifts in the elution profiles in the absence of molar mass information can easily result in misinterpretation of the data. For example: Initial MALS measurements in the absence of MgCl_2_ in the dark revealed an average molar mass, which was above the theoretical expected value for dimers indicating non-specific protein oligomerization. Upon light activation of the LOV domain, these higher oligomers dissociated, which was reflected by a decrease in the detected molar mass signal. However, the elution profile of the light activated protein shifted towards lower retention volumes compared to that observed in the dark. In the absence of molar mass information, such data could be easily interpreted as light-induced dimer stabilization or dimerization of *Pt*Au1a.

We carefully performed and repeated all MALS experiments and we are confident about the results presented here. We are aware of the fact that additional studies using complementary experimental techniques are required to investigate the interesting monomer-dimer equilibrium of *Pt*Au1a in more detail in the future. However, such studies clearly go beyond the scope of this manuscript.

We thank the reviewers for the suggestion to indicate the void volume of the gel-filtration column in the figures. We agree that this would be helpful for the reader and changed the figures accordingly.

*2) In the subsection “Blue light illumination induces dimerization of* Pt*Au1aLOV, but not of* Pt*Au1a_full_ and* Pt*Au1a_bZIP-LOV_”. The authors note a smaller than expected apparent MW for the* Pt*Au1a dimers. Such data is consistent with their H-D exchange data indicating a fast exchanging dimer. Indeed studies of the isolated LOV domain of* Pt*Au1a demonstrated fast exchange characteristics by SEC (Herman et al. Biochemistry 2015), where the elution volume and apparent MW were concentration dependent. Similar data was previously shown for* N. crassa *VVD (Zoltowski et al., Biochemistry 2008) and appears to be common in LOV proteins.*

*Did the authors examine the concentration dependence of the apparent MW by SEC? Alternatively the lower MW than expected could be due to the relatively fast photocycle (15-25 minutes).*

We also investigated the concentration dependence of the apparent molar mass of the bZIP domain containing *Pt*Au1a variants by SEC-MALS. We agree that this information might be interesting for readers of the manuscript and included chromatograms (Figure 2—figure supplement 1) of experiments performed for *Pt*Au1a_full_ and *Pt*Au1a_bZIP-LOV_ in the light at protein concentrations of 100 µM and 200 µM (the experiments shown in Figure 2 were performed at a protein concentration of 150 µM). For both protein variants the elution profile slightly shifts toward lower retention times with increasing protein concentration. Additionally, the detected average molar mass signal increased from 28.7 kDa (100 µM) to 35.1 kDa (200 µM) for *Pt*Au1a_bZIP-LOV_ and from 48.4 kDa (100 µM) to 54.4 kDa (200 µM) for *Pt*Au1a_full_.

As discussed above, we agree with the reviewers that the relatively “fast” photocycle of the bZIP domain containing variants might be one of the reasons why we did not detect strong stabilization of the *Pt*Au1a_full_ and *Pt*Au1a_bZIP-LOV_ dimers in our light experiments. We again want to point out that we could not detect indications for light-induced *Pt*Au1a_full_ and *Pt*Au1a_bZIP-LOV_ dimerization in any of our experiments.

*3) Figure 7: Why is there a kink in the SAXS density? According to the model this should be a symmetric dimer with a flexible linker.*

Symmetric dimers can have asymmetric structures; the myosin structure is an example. Nevertheless, we agree with the reviewers that the kink in the SAXS density obtained for the (*Pt*Au1a_bZIP-LOV)2_-DNA complex is surprising. However, this is the reproducible result of our reconstructions. Since we currently do not have structural information for the bZIP- LOV linker region, which could explain this result, we cannot give a clear answer to this question. We are working toward obtaining structural information for this region and hope that we can provide an explanation for this interesting observation in the future.